# An Overview of the Additive Manufacturing of Bast Fiber-Reinforced Composites and Envisaging Advancements Using the Patent Landscape

**DOI:** 10.3390/polym15224435

**Published:** 2023-11-16

**Authors:** Devarajan Balaji, Balasubramanian Arulmurugan, Venkateswaran Bhuvaneswari

**Affiliations:** Department of Mechanical Engineering, KPR Institute of Engineering and Technology, Coimbatore 641407, Tamilnadu, India; bhuvaneswari.v@kpriet.ac.in

**Keywords:** natural fibers, natural fiber-reinforced composite, patent landscape, additive manufacturing

## Abstract

Natural fiber composites attract attention owing to their environmentally friendly attributes. Many techniques, including fiber treatment, coatings, and fiber orientations, are used to improve the strength of natural fiber-reinforced composites. Still, the strength needs to be improved as expected. At present, some automation in manufacturing is also supported. Recently, additive manufacturing (AM) of natural fiber-reinforced composites has attracted many researchers around the globe. In this work, researchers’ attention to various natural fibers that are 3D printed is articulated and consolidated, and the future scope of the additive manufacturing of natural fiber-reinforced composite is envisaged using the patent landscape. In addition, some of the advancements in additive manufacturing of natural fiber composites are also discussed with reference to the patents filed lately. This may be helpful for the researchers working on AM of natural fiber composites for taking their research into new orientations.

## 1. Introduction

The global demand for renewable resources has led to the development of bio-composite materials as an alternative to conventional materials [1]. Natural fiber-based polymer composites are widely used in various technical applications such as automotive, aerospace, construction materials, electronic gadgets, and sporting goods [2,3]. The recent trend in developing new materials with enhanced thermal and mechanical properties involves the combination of natural fibers, nanofillers, and polymers, expanding their application potential possibilities [4]. Researchers are very interested in the possibility of improving the elasticity and toughness of natural fibers by incorporating clay or nano-clay elements into organic–inorganic hybrid nanocomposites. This exciting topic has potential uses in packaging food, medical equipment, and the automotive sector [5,6,7,8,9,10,11].

Even though modern materials like building materials, glass fiber/resin composites, metal, and acrylics are available as alternatives, there is still a lot of demand for clay-based goods because of their resilience, power, ability to insulate against heat and sound, and resistance to fire. Because of this, clay is still used as a basic building material by a lot of people [12,13,14,15]. New ways have been found to make clay-based building materials that work better and have better properties than those made with traditional firing methods [16]. Mixing is one of the best ways to make clay more stable. It involves mixing clay with materials like limestone, cement, and other garbage products. Sand is a porous material with poor tensile behavior and geological properties that change depending on the environment. This makes it hard to work with in areas where there is a lot of population expansion, development, unpredictability, and erosion. So, areas with sandy soils that are also sensitive to the environment need the right stabilization options [17,18,19,20,21,22,23,24]. Multiple methods have been found to settle uncertain soils prior to being used for building or other growth.

Using 3D printing to create composite materials of varying shapes and sizes, along with dimensions from natural materials and clays, has gained popularity in recent years. The ease of implementation, effectiveness, and low cost of this method have all contributed to its rise to prominence. Improved material quality and performance have resulted from the use of additive manufacturing in the fabrication of a number of polymeric composites. These include filters, membranes, and metal–organic frameworks (MOFs). Particularly effective in enhancing water quality, 3D-printed filters for sewage treatment stand out [25,26,27,28]. The potential for creative uses is expanding due to the advent of 3D printing. Material accessibility, manufacturing pace, and generating resolution are just a few of the factors that must be taken into account when implementing the technology of additive manufacturing to a particular use case [29].

It is essential to keep in mind that 3D printing is not meant to be used in isolation, but rather as part of a more extensive system involving multiple processes or an integrated strategy that makes use of a number of different methods [30,31,32]. This makes it possible to develop novel materials and meet emerging product needs. Research into the printability of various materials, such as fast-drying concrete, smart materials, biomaterials, nano-materials, and functional materials, has increased in response to the rising demand for complex and multifunctional products [33,34,35]. Ceramics and concrete, for example, present difficulties for 3D printing because it is impossible to fuse powders using only heat at their melting temperature. However, by increasing the boiling point or glass transition temperature, polymers and metals might become fused [36,37]. The process of additive manufacturing in ceramics presents unique challenges due to the material’s substantially greater point of melting metals and polymers [38,39,40].

The printability and mechanical characteristics of a 3D-printed mortar containing copper tailings from industrial waste were studied by Ma et al. [41]. They found that adding copper tailings up to an optimal level of 30% increased the material’s constructability and compressive strength. Because of their many benefits over synthetic fibers, such as lower production costs, fewer negative effects on the environment, greater biodegradability, and longer lifespan, natural fibers are increasingly being used in cementitious composites. Natural fibers have a number of advantages over synthetic fibers, including glass and carbon, including being biodegradable, lightweight, less dense, and non-toxic and having a higher strength-to-weight ratio. Furthermore, natural fibers demand less energy to produce and process than synthetic fibers. Compared with the energy needed for the manufacture of a comparable quantity of polypropylene synthetic fibers, the amount of energy required for the manufacturing of natural jute fibers is roughly 7%. When included in cement composites, natural fibers may drastically change the characteristics of cementitious materials due to their hydrophobic nature. In order to maximize fiber utilization and boost the characteristics of cementitious composites, these constraints must be removed [42,43]. In addition, fiber reinforcement can significantly improve the properties of polymer matrix substances.

For the creation of fiber-reinforced polymer composites, FDM and direct writing have been widely used in 3D printing technology. Blended polymer pellets with fibers are then fed into an extruder to produce filaments for FDM. An additional extrusion method might be used to ensure that the fibers are distributed evenly. Polymer paste and fibers have been combined using the direct writing method and then extruded. It is difficult to get a uniform layer of powder and fibers when using powder-based technology, so they are not permitted to make fiber-reinforced composites [44,45]. Additive manufacturing methods for natural fiber and polymer composites have been the subject of multiple recent research efforts. Many investigations have focused on the molding and manufacturing techniques, modification strategies, and mechanical characteristics of biomass composites. Most polymer matrices are hydrophobic, whereas natural fibers are hydrophilic; this polarity mismatch makes bonding between the two materials difficult. Improving the interfacial bonding by altering the surface is a common practice. Mechanical and interfacial characteristics might be enhanced by optimizing the polymer matrix and fiber content [46].

Fibers act as nucleating agents, speeding up the crystallization procedure during polymer mixture treatment. Particularly for PLA, which displays slow crystallizing and poor crystallizing capability under substantial cooling rates, enhancing the crystallization rate becomes essential for 3D printing implications. These features slowly cure and mold following melt extrusion printing. To combat this issue, many methods are being used to boost the efficacy of both the matrix and fibers from nature. Chemically modifying the fiber surface, increasing fiber size, adding the right amount of fiber and modifier, and controlling the interface between the fibers and the matrix are all examples of these methods [47]. It is evident from the literature that additive manufacturing is now widely used across various industries, which was previously considered challenging. This article aims to highlight this key feature and consolidate the utilization of additive manufacturing in printing various natural fibers while also exploring the potential scope of this technique. Alongside this, a patent landscape analysis is coupled with the current literature survey to identify the gap in AM techniques for the printing of natural fiber-based composites, which has not been dealt with in much of the literature lately. This paves the way in integrating the currently used technology with the latest advancements in the field, which will help researchers working in this field to use these advanced techniques for begetting numerous state-of-the-art applications of 3D printed components.

## 2. Methodology Used to Print Natural Fibers

Printing natural fibers consists of three basic methods: particle printing, fiber printing, and nanoprinting. In order to be printed, the fibers need to be converted into filament form. The most commonly used printing technique is fused deposition modeling (FDM), which is favored due to its cost-effectiveness compared with other additive manufacturing methods. Figure 1 illustrates the different forms of fibers combined with various polymeric materials such as polylactic acid (PLA), acrylonitrile butadiene styrene (ABS), and others.

## 3. Natural Fibers Printed Using the Additive Manufacturing Technique

### 3.1. Banana Fiber

Banana fibers (BFs) are chosen as a reinforcement material because of their biodegradability and biocompatibility, and recycled ABS along with polyamide 6 (PA6) were used as the primary matrix materials in some of the research works. In some studies, ABS with PA6 and BF were mixed using a standard twin-screw extrusion (TSE) technique, and the resulting feedstock filament wire was then used in 3D printing. This BF-reinforced ABS with PA6 filament was successfully used in an unmodified publicly accessible FDM 3D printer. The objective was to print functional prototypes with enhanced thermal, mechanical, and morphological properties. Thermographs and micrographs, along with mechanical analysis, all supported the obtained results. Figure 2 shows the tensile specimen of ABS with 5% BF [48].

Using fused filament fabrication (FFF) and in situ impregnation, another research explored the possibility of using BF as a continuous reinforcement with a thermoplastic matrix to form a composite. Microstructural defects, fiber eccentricity, and porosity were examined as a function of printing speed along with conditions at a central composite design test. The microstructure of the specimens was investigated using optical and stereo microscopy, and the porosity and fiber eccentricity were calculated using image analysis. The results showed that the eccentricity of the fibers within the matrix was unaffected by the experimental factors. However, the porosity level was found to be affected by the printing speed. Printing at 321 mm/min instead of 180 mm/min reduced the porosity from 9.5% to 0.3%. The wet conditions of the fiber embedded over a thermoplastic matrix were found to be responsible for this behavior [49].

A few research works aimed to examine the impact of FFF settings on the quality of printed parts when applied over composite material filament. A filament from BF and PLA has been used in some studies. This composite filament fabricated samples with varying layer thickness, infiltration, and build orientation. Standardized procedures for flexural and tensile testing were followed. Infill percentage and build orientation had significant effects on the investigated material characteristics. The ‘on edge’ construct orientation had the best material characteristics of the three. Tensile strength was 73% and 77% greater in the ‘flat’ and ‘on edge’ orientations, respectively, compared with the upright position. Furthermore, the ‘flat’ and ‘on edge’ orientations both had superior flexural strength to the upright position by a factor of 60% and 70%, respectively [50].

### 3.2. Kenaf Fiber

Specimens were 3D printed after being prepared with a TSE machine using fiberglass-reinforced ABS composites of varying volume percentages. Tensile and flexural tests and morphological analysis were performed on the samples. The following is a synopsis of the study’s results: The tensile strength and modulus of pure ABS were measured at 23 MPa and 328 MPa, respectively. In contrast, the composites made using FDM saw a reduction both in tensile strength and modulus after kenaf fiber was added. The 2.5% ABS-kenaf composite had the highest strength and modulus values of 21.7 MPa and 321.7 MPa, respectively, when compared with ABS composites containing 5%, 7.5%, and 10% of kenaf fibers. The tensile strength and modulus of the 5% kenaf-ABS composite, and the modulus of the 5% ABS-kenaf composite were 11.5 MPa and 184.5 MPa, respectively; these values were the lowest of all tested. Poor fiber–matrix interfacial adherence, strand pull-out, fiber breakage, and porosity levels were all revealed with a morphological evaluation of the tensile fractured samples. The flexural strength and modulus of pure ABS were measured at 40.6 MPa and 113 MPa, respectively. The composites’ flexural strength and modulus decreased, however, after kenaf fiber was added. The flexural strength was greatest for the 2.5% ABS-kenaf composite (33 MPa), while the flexural modulus was greatest for the 10% ABS-kenaf composite (88.5 MPa). The flexural strength along with modulus of the 5% ABS-kenaf composite were the lowest of all tested materials. The deterioration phenomena and the emergence of porosities were found in the morphological analysis of the flexural fractured samples. In summary, the ABS-kenaf composite demonstrated promise for manufacturing using FDM technology, despite slightly lower tensile and flexural testing findings compared with pure ABS. By testing tensile and flexural characteristics, the researchers of this study confirmed that filaments made from ABS polymer composites reinforced with kenaf fiber are suitable for FDM machines. These characteristics are frequently used as benchmarks of technical product performance, so they play an important role in the materials and design industry. Therefore, studying and comprehending these characteristics is crucial. Analysis of the composites’ morphology shed light on their mechanical properties, showing that a lack of interface bonding among the matrix and fiber was responsible for the decline in these characteristics. Evidence for this was seen in events like fiber pull-out, the development of porosity, and fiber breakage, all of which pointed to fewer bonds among matrix polymer molecules and fiber. Figure 3 shows the process of making the ABS with kenaf filament and testing them [51].

Some experimental works examined the tensile performance of 3D printed composites along with the physical and thermal characteristics of a natural composite filament used in 3D printing to establish its viability as an element in ankle–foot orthosis (AFO) usage. SEM examination, thermogravimetric or differential scanning calorimetry investigation, and an analysis of the filament morphology following extrusion with varying fiber loadings were just some of the physical tests performed. Printed samples with 0, 3, 5, and 7% fiber loads were subjected to tensile tests. The results showed that strength increased with fiber load to a maximum of 3% but dropped significantly at 5 and 7 wt. % owing to the existence of voids. The results also showed that the filament structure was significantly affected by the extrusion temperature, which in turn affected the mechanical properties of the printed composite materials. These findings suggest that using kenaf—PLA coil filaments in printing AFOs is possible, provided the filament fabrication and printing processes are subject to adequate oversight. This work is unique in exploring the potential of kenaf—PLA coil filaments for 3D printing, particularly in the context of making AFOs [52].

Researchers have spent a lot of time trying to figure out the best way to use natural fibers for support in high-performance materials. There are a number of ways to incorporate these reinforcing agents into a polymer matrix. The physical and mechanical characteristics of composites made by mixing kenaf fiber with thermoplastic PLA has been investigated. Faster production, better finishing, a capacity to produce complicated forms with intricate shapes and dimensions, and lower costs make FDM stand out among the different additive technologies. The purpose of the research was to examine the results of extensive mechanical and physical tests on kenaf fiber to determine the effects of chemical treatment. The goal of the study was to develop a filament for 3D printing by combining kenaf fiber and PLA polymer at a 2.5 wt. % concentration using a thermoplastic melt extruder. An untreated kenaf fiber composite material, a polymer, and three variations of a silane treatment were all created as test conditions. The specimens shown in Figure 4 were printed to ASTM specifications and were put through rigorous testing, providing useful information. The results showed that increasing the concentration of silane by 1% and then treating the mixture with a 6% alkali solution improved the interaction between each of the phases. Removing the natural fiber’s lignin, hemicellulose, and cellulose, which are three chemical components, was responsible for the improvement. The research also showed that fiber destruction can occur at higher silane levels. For example, the 2% silane level was the weakest of all those tested, and it was inferior to the 1% concentration. On the other hand, untreated composites made from natural fibers had the weakest surface shear strength because of insufficient interfacial bonding. Evidently, the development of exceptionally strong functions relied on finding the ideal amount of silane for the treatment of the surface when utilizing altered composites including natural fibers [53].

### 3.3. Flax Fiber

Separate pathways were used for integrating continuous twisted yarn along with polyamide 6 (PA6) into the altered FFF process in a research work. Within the heated area of the nozzle, the two components were combined, and the fibers were impregnated. The benefits of this technique include (i) the ability to impregnate and print at the same time, (ii) the ability to tailor the quantity of fiber by modifying the process settings or nozzle diameter (a feature that is still in the works), and (iii) the ability to prevent fiber degradation by reducing the amount of time the fiber spends in the nozzle. The following is a synopsis of the research results: The composites’ tensile modulus was increased by a factor of 9 and their strength by a factor of 2.4, in contrast with pure PA6, thanks to the novel approach used in the study. Microstructural heterogeneities were reduced, for the yarns with volume fractions of 18% and 22%, by using the lowest linear weight bleached textile flax yarns available (26 Tex). Tensile characteristics of continuous fabric flax fiber in PA6 composite materials were studied, with an emphasis on the role played by the volume fiber ratio and fiber orientation. The weakest feature of the studied flax-reinforced composites was the fact that the tensile characteristics increased with enhancing volume fiber ratio but decreased dramatically when a transversal or 45° fiber orientation was chosen [54].

Non-linear tensile behavior was observed in the unidirectional fabric flax with PA6 composites, typical of natural fiber one-directional composites made using traditional techniques. A marked reduction in void content and an inter-layer breakdown in PA6 composites was observed with increasing volume fraction (Vf) on one-directional fabric flax. The Rule of Mixture was used to verify the longitudinal elastic and strength characteristics, and it was found to be in perfect accordance with experimental data. Using flax fibers, and more specifically, twisted flax yarns, as reinforcement in composites causes problems due to the incompatibility between the fibers and the matrix, which results in weaker fiber/matrix interfacial adherence. Because debonding and pore formation are encouraged under these circumstances, the strength characteristics of the composite materials are diminished. Therefore, additionally, mechanical property enhancements necessitate extra physical or chemical alterations. The research was conducted to determine if the proposed FFF process modification could successfully apply to the recently developed composite material. Improving the quality of flax fiber yarn and its filaments and enhancing the Vf (until 33%) are two areas where future research could help improve the mechanical attributes of printed continuous clothing flax with PA6 composites. Long-term performance (which incorporates fatigue assessment) and robustness (which incorporates impact assessment) of the manufactured composites should also be investigated. Figure 5 shows the modified version of the 3D printer used to print flax with PA6 [54].

While PLA has found extensive application in additive manufacturing, its weak mechanical characteristics limit its usefulness in other contexts. To overcome this constraint, composite materials often include fibers to improve their properties. The increase in tensile and flexural strength was achieved by incorporating flax fibers into the development of PLA composites in some research works. Using an L18 Taguchi array, the authors varied the extruder temperature (at three levels), the number of strands (at three levels), the infill proportion of the specimens (at three levels), and the chemical process used on the flax fibers’ surfaces. Three-dimensional printing equipment was used to create tensile and flexural samples, with a custom mold created to fix and correspond to the fiber strands throughout printing. Both the tensile and flexural tests were carried out in accordance with international standards (ASTM D638.14 [55] and ISO 14125 [56], correspondingly). The results showed that the exterior treatment of fibers with chemicals (NaOH) had little effect on the composites’ mechanical characteristics. The infill density, nevertheless, was found to have a significant effect on the improvement in mechanical ability. The maximum tensile stress was 50 MPa, and the maximum bending stress was 73 MPa. The mechanical characteristics of PLA composites were successfully enhanced with the addition of natural fiber reinforcement [57].

Parts made from continuous flax fiber-reinforced plastic (CFFRP) can now be printed using a novel technique with a five-axis three-dimensional printer that utilizes FFF technology. The stair-step effect and inferior mechanical characteristics are commonplace in conventional FFF printing because of the requirement for frames to support it. To address these constraints, some studies made arched routes for the model and G-code out of continuous prepreg filaments made from natural fibers. The surface quality of the CFFRP components was greatly enhanced by printing them with a five-axis 3D printer. The CFFRP composite had a significantly higher tensile strength (89%) and modulus (73% rising) than the PLA filaments. In a comparable manner, the flexural strength (211% rising) and modulus (224% rising) of the CFFRP samples printed using the 3D printer were significantly higher than those of the PLA samples. In contrast to flat slicing, the CFFRP samples printed using the 3D printer displayed a 39% rise in the bent force load and an incredible 115% enhancement in stiffness when using the highest possible arched bent force load and stiffness. These composites show great promise for use in the automotive industry due to their favorable characteristics, which allow for the creation and manufacturing of highly sophisticated lightweight components like leaf springs. This novel strategy expands the potential uses of CFFRP composites in the automotive industry. Figure 6 shows the PLA flax composite filament-making device [58].

### 3.4. Hemp Fiber

By mixing pre-consumer polypropylene that was recycled using different quantities of hemp or harakeke material during the extrusion process, composite filaments were created. FDM was employed to create tensile test samples from these filaments. Tensile testing was performed on the filaments along with the test samples, and the results were compared to those of plain polypropylene samples to determine the differences in properties. Ultimate tensile strength and Young’s modulus of the reinforced strands, were significantly higher than those of the polypropylene filament, increasing by more than 50% and 143%, correspondingly, for either 30 wt. % of hemp or harakeke [59]. However, not all FDM test samples showed the same degree of improvement: some had properties even worse than unfilled polypropylene. SEM examination of the fractured areas showed reasonably aligned fibers with uniform distribution and also revealed an abundance of porosity and fiber pull-out. Despite these difficulties, fiber reinforcement was discovered to be beneficial in maintaining dimensional stability throughout extrusion and FDM, which is essential for effective execution in printing with FDM. The researchers provided suggestions for improving the appearance of the printed frameworks and their mechanical characteristics by boosting the conditions of printing. The functionality of the printed samples can be impacted by issues like porosity and fiber pull-out, which are targeted using these optimizations. It is hoped that by adjusting the settings for the process, the FDM-printed composites’ quality of construction and mechanical characteristics, can be enhanced even further [59]. The tensile specimen of polypropylene with various weight percentages of hemp is shown in Figure 7.

One research work looked into the viability of a new 3D printable material that combines silicone with hemp fibers made from sustainable and environmentally friendly resources to improve material mechanical characteristics. The incorporation of fibers enhanced the mechanical characteristics of the silicone matrix. Nevertheless, the material’s high viscosity made it challenging to print with silicone. To remedy this problem, different solvents were added to the mix to adjust viscosity and boost printability. The primary goal of the study is to improve composite printing technology by identifying the optimal combination of hemp fibers, silicone, and a solvent for creating a printable material. Rheological studies were conducted on the recently engineered material to determine the best printable formulation. The mechanical properties were enhanced while printability was maintained in a composition containing 15% hemp fibers and 20% solvent. The specimens from both 3D printing and molding were tested for their mechanical qualities as well. According to the findings, the 3D-printed samples performed better than their molded counterparts. The developed material was then successfully used to create a honeycomb structure and a simple gripper. The printability and mechanical characteristics of a silicone–hemp fiber composite was investigated with the aim of expanding the material’s potential uses. The results help pave the way for more environmentally friendly, high-performance 3D printing materials [60].

In the field of additive manufacturing (AM), FFF is a common technique. In another research study, the constraints of a pure PLA filament were addressed by introducing and blending it with hemp fiber, an ecologically friendly and biodegradable material, at varying weight percentages (3%, 7.5%, and 10%). Fatigue analysis of pure PLA and several hemp-fiber-infused PLA specimens showed that, up to a specific weight percentage, the infused PLA specimen improved its maximum bending stress and a whole fatigue life over pure PLA. When compared with pure PLA, the average ultimate flexural strength of specimens containing 10 wt. % of hemp fiber was 7.32% higher. The addition of hemp fiber resulted in an increase in the average Young’s modulus of 10.7% for the specimen containing 10% hemp fiber and an increase of 23% for the specimen containing 7.5 wt. % hemp fiber compared with PLA. Furthermore, when compared with PLA, the fatigue life of the specimens made with 10% weight of hemp fiber was 4.05% higher. However, the study found no statistically significant benefits from adding 3% weight of hemp fiber. These results shed light on AM processes and aid in the creation of eco-friendly composites for use in a wide range of sectors. The study showed that the mechanical characteristics of PLA could be improved, and feasible additive manufacturing practices could be promoted by including hemp fiber-infused PLA filament [61].

### 3.5. Jute Fiber

Various studies were conducted on the 3D printing of jute fiber-reinforced PLA composites. Dog-bone tensile samples (ASTM D638) were effectively printed after a PLA-fused filament was placed onto jute fabrics for characterization of the composites. The mechanical characteristics of PLA with jute textile composite materials were enhanced with chemical enhancement, treated with flame retardant additives, and sprayed with aerosol adhesive. The elastic modulus and strength were shown to be greater for pure PLA than for PLA composites, whereas plastic deformation was found to be somewhat lower for PLA composites. Tomography scans showed that the fabrics were well-oriented and that the jute fabrics adhered to the PLA to some degree. While there was no change in the glass transition temperature, the viscoelastic characteristics of the PLA composites lowered the storage modulus and moderated the damping factor due to segmental motions. Jute fabrics treated with flame retardant and spray adhesive outperformed PLA and PLA with altered fibers in terms of their resistance to combustion. These results emphasize the need for additional thorough research into the consequences of using plant fiber materials as reinforcement in objects produced using 3D printing with applications in industry. Jute fiber fabrics were shown to have promising potential as a reinforcement choice in PLA composites, and the investigation highlights the importance of investigating their efficacy in targeted industrial settings [62].

Another study presented a manufacturing process for 3D-printed parts using jute-reinforced PLA filament obtained from 5% waste jute. To improve the bonding between materials in the composite matrix structure, the authors milled PLA granules to increase the surface area of the material. The effectiveness of this milling technique was demonstrated by comparing the production of the same composite matrix using unmilled PLA granules. Both matrices were then converted into filaments for 3D parts using fused filament manufacturing techniques. Thermogravimetric analysis (TGA) and differential scanning calorimetry (DSC) results were presented in filament format and provided information on filament properties. Additionally, the mechanical properties of 3D-printed parts were discussed. The primary purpose of the study was to investigate the effect of material size on the production of natural fiber-reinforced filaments for additive manufacturing. Researching this aspect provided valuable insight into the feasibility and potential benefits of using jute-reinforced PLA filaments in additive manufacturing processes [63].

The use of FDM for the three-dimensional printing of continuously printed fiber-reinforced thermoplastics has been suggested as a novel approach. This method may become the de facto standard for future generations of composite production as it provides a direct 3D production approach without molds. Thermoplastic filaments and continuous fibers are fed into a 3D printer independently, and the fibers are impregnated by the thermoplastic filament throughout the heated nozzle right before printing begins. PLA was used as the matrix substance in some experimental works, and the reinforcements included both carbon fibers and swirled yarns made from organic jute fibers, as shown in Figure 8. Examples of plant-sourced composites included those made by reinforcing thermoplastics with single-directional jute fibers. In contrast, composites made using the same method with single-directional carbon fibers outperformed both the jute-reinforced and unreinforced thermoplastics in terms of mechanical characteristics. The printed composites had greater tensile strength than regular 3D-printed polymer-based composites because of the integration of continuous fiber reinforcement. Using this novel strategy, 3D-printed composite materials can achieve higher functionality and efficiency [64].

### 3.6. Ramie Fiber

Biocomposites were created using the in situ impregnated FDM method, with dry-swirled continuous ramie fibers serving as the reinforcing phase and PLA serving as the matrix. The authors used single axial tensile and peeling examinations to evaluate the ramie fiber-reinforced biocomposites (RFRBCs) produced with varying processing variables to determine their mechanical properties and interlayer endurance. Prior to and following mechanical testing, the biocomposites’ morphology and characteristics were examined to learn more about the multiscale interfaces at play. The results showed that the biocomposites’ mechanical techniques depended on the interfacial characteristics between the formed layers, the ramie yarn with the matrix, and the ramie fiber with the matrix. Changing the ability of the matrix to flow, establishing pressure among the printing nozzle and a bed, and the impregnation period for ramie fibers were examples of how printing variation in parameters may result in substantial microstructure alters in the biocomposites under study. The optimal printing conditions for producing biocomposites with a tensile force of 86 MPa, minimal porosity, and a favorable interaction among deposited lines, and the PLA matrix along with ramie fibers was a printing process temperature of 220 °C, a layer thickness of 0.3 mm, and a printing rate of 100 mm per min. For producing biocomposites with acceptable mechanical attributes and enhanced interfacial qualities, these results show the significance of optimizing printing variables. Figure 9 shows ramie with PLA filament testing and analysis [65]. The fiber bundle interface and its morphology has been illustrated through red dotted lines.

A live-impregnated AM technique was used to create buildings, with the printing paths being either novel or conventional (one-directional or orthogonal). This research study used the quasi-static penetration test (QSPT) to analyze how 3D printing design, encouragement span for indenter diameter ratios (SIRs), and fiber reinforcement affected the penetration characteristics of biocomposites. Furthermore, the backlight technique was used to capture the damage procedure of the biocomposites in the real-time QSPT. Adding continuous ramie yarn and decreasing the SIR led to an increase in the penetration property and the energy absorption capacity of the 3D-printed specimens. Three-dimensional-printed biocomposites that had a woven-like architecture showed 31% greater energy absorption and 18% greater maximum penetration force than those with a non-woven-like design (one-directional) at an SIR of 5. The connection between the structure and the penetration characteristics of the three-dimensionally printed woven-like designed biocomposites was demonstrated with an examination of multiscale failure features and penetration damage techniques. Specifically, the beneficial effects of particular designs and fiber reinforcement for improving energy absorption and resistance to penetration were brought to light, providing important perspectives into the efficiency of biocomposites and their behavior under penetration forces [66].

Continuous ramie fibers, which are used in 3D printing and are renowned for their exceptional shape memory effects, have been examined to determine how they behave under quasi-static compression. Energy absorption (EA) and compression behaviors of CFCSs were investigated by subjecting them to in-plane compression tests and varying cell designs, the strand volume fraction (Vf), and fiber addition. Incorporating continuous ramie yarn and decreasing the V_f_ were found to enhance the compression characteristics and EA ability of the 3D-printed CFCSs. When compared with other cell shapes, the capacity for compression and particular EAs of the CFCSs with reversed trapezoid cell shapes (CFRTCSs) were the best. The in-plane compression resistance of CFRTCSs was predicted using an analytical model, and the results agreed well with measurement results from experiments. In addition, the idea of using composite frameworks reinforced with continuous natural fibers for shape memory functions was introduced in the study. The results of the shape-recovery tests showed that the 3D-printed CFCSs had great potential as essential parts in compact programmable smart systems. These results highlight the benefits of particular cell designs, fiber reinforcement, and the potential usages for these frameworks in shape memory technology. The results also contribute to our comprehension of the compression response and absorbing energy capacities of 3D-printed CFCSs [67]. Table 1 consolidates the mechanical properties of 3D-printed bast-fiber reinforced biocomposites.

## 4. Potential Research Gap

After studying the best settings for 3D printing with natural fiber reinforcements, it was found that there is a need for more study into identifying and optimizing processing variables unique to 3D printing natural fiber-reinforced composites, despite the fact that methods for additive manufacturing are being investigated for composites made from natural fibers and polymers. The printability, interfacial adhesion, and mechanical characteristics of the printed composites are investigated by varying parameters, including printing temperature, extrusion pace, layer height, and the orientation of the fibers. The creation of environmentally friendly methods of surface modification for natural fibers enhances the interfacial adhesion of natural fibers with hydrophobic polymer matrices. However, many of the current surface modification techniques involve the application of chemicals that could be harmful to the environment. The analysis of sustainable and environmentally friendly methods for surface modification that can efficiently improve bonding between surfaces with minimal negative effects on the environment is understudied. The slow crystallization kinetics of PLA in 3D printing has prompted the exploration of novel approaches to speed up the curing and forming of printed parts. While a number of approaches have been used to address this problem, the crystallization process for PLA in 3D printing could benefit from more study into novel strategies. In order to speed up crystallization without sacrificing other material properties, it may be necessary to experiment with different nucleating agents, fine-tune the composite formulation, or add additives. Dispersion of natural fibers within the polymer matrices has also been an ongoing issue while manufacturing biocomposites. AM provides a clear solution for dispersion by predefining the orientation of fibers and particles within the thermoplastic matrix. However, a better interlayer fusion is required to avoid delamination and to increase the strength of 3D-printed composites. By filling in these knowledge gaps, 3D printing of natural fiber-reinforced composites could be advanced, the bonding between fibers and polymers could be strengthened, and the processing, dispersion, and mechanical characteristics of printed parts could be enhanced. Due to this knowledge gap, scientists have begun to consider the application of this method. To explore the potential reach of this method into the future, this article also surveys the patent landscape.

## 5. Patent Landscape

The range of natural fiber composites printed using additive manufacturing was evaluated using patent status. Table 2 below shows an English search for natural fiber additive manufacturing in all categories using the keywords “additive manufacturing” and “natural fiber”. Other valid criteria stemmed from searching for related keywords. International Patent Classification (IPC) is a classification system that corresponds to the listing of patents filed in a specific subfield. In this case, the IPC denotes the additive manufacturing of natural fiber composites with specific natural fibers. Also, only one patent is considered if the same patent is filed in different countries. The total number of patents pending is 684. The ‘counts’ column next to every classifier column including countries, IPC, and year denotes the number of patents filed in each of the preceding columns.

## 6. Advancements in AM of Biocomposites

The patent landscape reveals that few techniques may grow better. It is assessed based on its potential novelty and consolidated to provide future directions for the additive manufacturing of natural fibers for the researchers working in this field. The following subsections deal with some recent advancements in AM of natural fiber composites. All the following studies are patented and were published during last 10 years.

### 6.1. AM of Natural Fiber-Reinforced Bioplastics

Apart from conventional polymers, recently, bioplastic matrices were also used during the AM of natural fiber composites. The plasticization of a binder in a first extruder, where the said binder is a polymer, is one step in the process of making an aloe vera-reinforced biocomposite (ARBC) material. During the method a hydrophobic agent, either dispersed or dissolved in a liquid carrier, is included along with a fibrous filler material derived from aloe vera, specifically, aloe vera leaves, and combining the dried blend alongside the plasticized binder after it is sheared and dried mechanically in a second extruder, in which fluid is at least partially or entirely taken out of the mixture. This method results in the creation of ARBC material, which is then utilized in the container manufacturing process. This method is used for the production of bioplastic composite filaments, granules, or pellets for additive manufacturing and bioplastic composites with antiseptic characteristics and UV-light absorbing characteristics [69]. The stepwise procedure for producing bioplastic filaments and its outcome is shown in Figure 10.

### 6.2. Usage of Dual Natural Fiber

The need for finding materials and techniques that improve the deterioration of a 3D-printed object has been recognized by researchers. The addition of plant-based fibers, in particular castor oil plant fiber and pineapple leaf fiber, can improve the degradation property of the resulting objects, so it is important to find plant-based compositions that may be utilized for producing them using FDM. Finding ways to make and use compositions based on natural fibers is also necessary. One study described the method of creating a 3D object in a composition using the FDM additive manufacturing method. A combination of a thermoplastic polymer and two natural fibers, each with a different percentage contribution, could shorten the time it takes for an object to degrade in comparison with one made entirely of thermoplastic polymer. The combined volume fraction of both the natural fibers was 6%, while the thermoplastic polymer matrix occupied the remaining volume. The degradation time of a three-dimensional object made with produced natural fibers and PLA was 40% less than that of an identical object made with only PLA, and its degradation rate was 40% higher [70].

### 6.3. Three-Dimensional Printing of Binder-Embedded Natural Fiber

The process for FDM-based AM of a preform, including the steps for providing an uninterrupted filament that includes a natural fiber bundle as its inner core and a binder coating that packs the filament entirely has been described in some studies. Feeding the filament into an extrusion machine at a feed rate and heating the filament to a temperature that depends on the compromising temperature associated with the binding agents and the filament melting point is also described. The derived preform and the reinforced composite part that can be made from it are detailed in Figure 11 [71].

Another study included instructions on how to make composite feedstocks from modified natural fibers. Agricultural fiber material, alternatively soybean hulls, are hydrolyzed under conditions for a period of time adequate to eliminate some or all of the arabinose that is present in the agricultural fiber material, resulting in arabinose-deficient hydrolyzed goods. The arabinose-deficient hydrolyzed goods are then hydrolyzed according to circumstances for a long enough time to eliminate some or all of the xylose in the arabinose-deficient hydrolyzed product. A method for 3D printing a structure out of the modified fiber composites is additionally offered, as are techniques for enhancing a minimum of one property of the modified thermoplastic copolyester composites along with techniques for enhancing FFF processes and for isolating xylose eliminated from arabinose-deficient hydrolysates [72].

### 6.4. Three-Dimensional Printing of Multiple Fibers

A newly developed approach for the 3D printing of renewable materials like natural fibers, mineral fibers, and plastic fibers to manufacture composite parts has emerged, wherein a fluid matrix substance is placed into a production device in stages, one after the other, to create an additively manufactured part. At least some of the liquid matrix material and/or the reinforcing element is incorporated into the fluid matrix. In addition to being cheap and safe to use around the body, natural fibers also have high-quality mechanical characteristics. Using materials such as natural fibers improves the durability of composite parts made using the disclosed method and makes their disposal and recycling easier. In a broader sense, biopolymer fibers are also considered natural fibers and may be utilized in the process described herein. Although mineral fibers are typically more challenging to process, they have excellent mechanical properties. Mechanical durability and processability are both enhanced with the use of carbon or aramid fibers. When the mechanical characteristics of a composite part must meet specific requirements, it may be practical to use a combination of the materials mentioned. Combining carbon and aramid fibers, for instance, results in both high strength and excellent impact toughness, thanks to the respective properties of the two types of fibers [73,74]. Synthetic or natural fibers can be used as reinforcing components in additive manufacturing compositions. Small yarn and a thermosetting resin make up most polymer composites. Carbon fibers, basalt fibers, aramid fibers, natural fibers, glass fibers, and thermosetting resins are all viable reinforcement options. Reinforcing materials can take the form of either a continuous fiber ejected alongside the thermosetting substance or a distribution of discontinuous fibers, such as those made from aramid carbon, or glass. Two or more of the items mentioned above can be used in the reinforcement [75,76].

## 7. Conclusions

This work articulated and consolidated to gain the researchers’ attention on various natural fibers that are 3D printed. In addition, the future scope of additive manufacturing of natural fiber-reinforced composite was also envisaged using the patent landscape. Natural fiber composites printed using the additive manufacturing technique were found to have potential growth with respect to novel properties. In the patent landscape analysis, the USA was found to be dominant because of the developed end products in their country, thereby considerably reducing imports. Some of the potential inventions address the identified research gap. Some thermoplastic materials other than PLA and ABS were also experimented with using a dual extrusion setup. The utilization of dual natural fibers and natural fiber bundles embedded in PLA filaments was also patented, which contributes to hybrid composite manufacturing using AM techniques. Recent patents have also dealt with the use of multiple classes of fibers for the manufacturing of biocomposites using 3D printing, which opens up various avenues in different engineering applications such as biomedical, healthcare, and construction applications. From this article, embedded with patent landscape analysis, researchers are expected to obtain guidelines to proceed further in the domain of additive manufacturing.

## Figures and Tables

**Figure 1 polymers-15-04435-f001:**
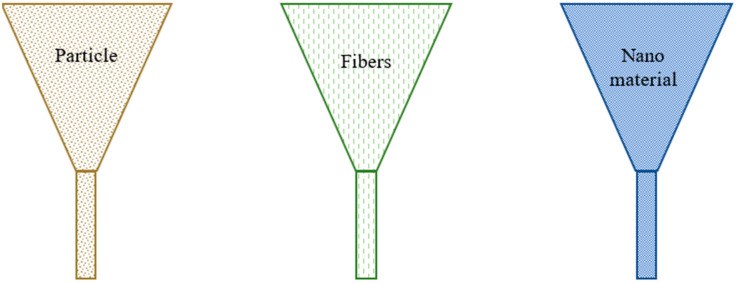
A 3D printer printing head with various forms of natural fibers being mixed with other polymeric materials and being printed.

**Figure 2 polymers-15-04435-f002:**
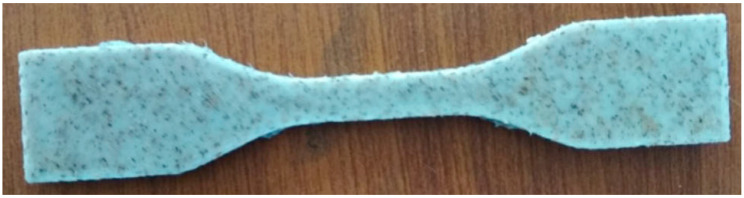
Tensile test specimen—ABS-5% banana fiber composite [48].

**Figure 3 polymers-15-04435-f003:**
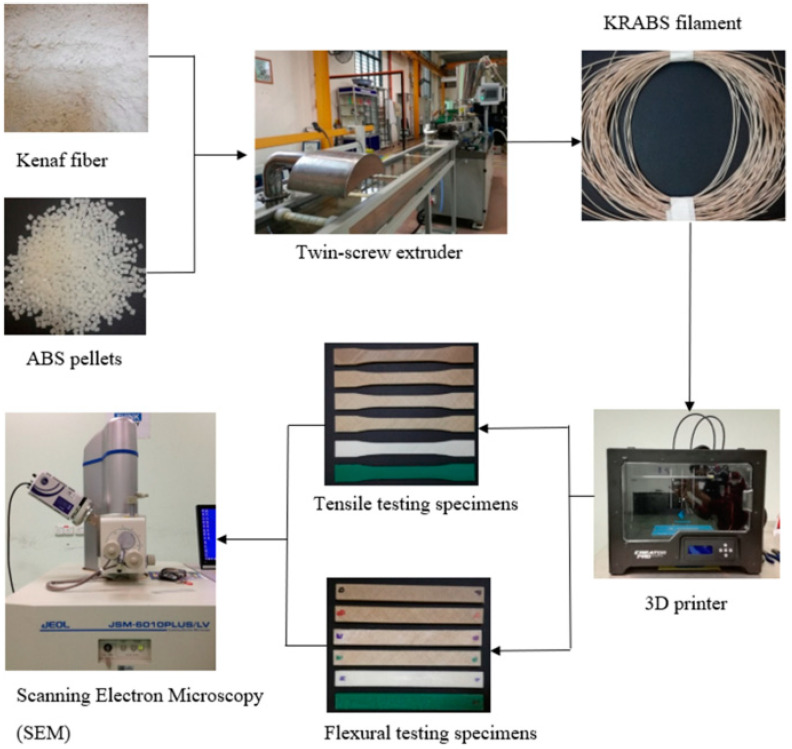
ABS with kenaf fiber fabrication and printing process [51].

**Figure 4 polymers-15-04435-f004:**
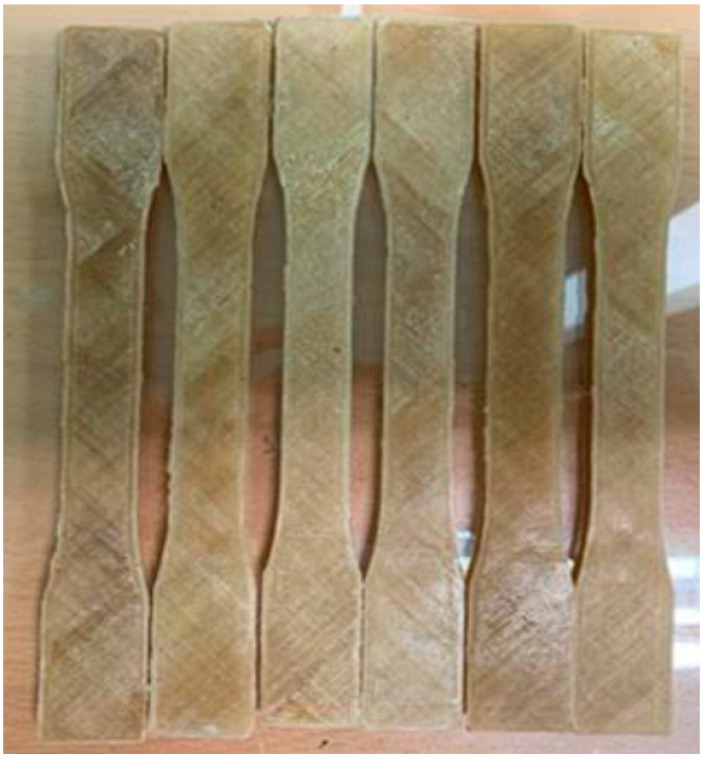
PLA with kenaf fiber tensile specimen [53].

**Figure 5 polymers-15-04435-f005:**
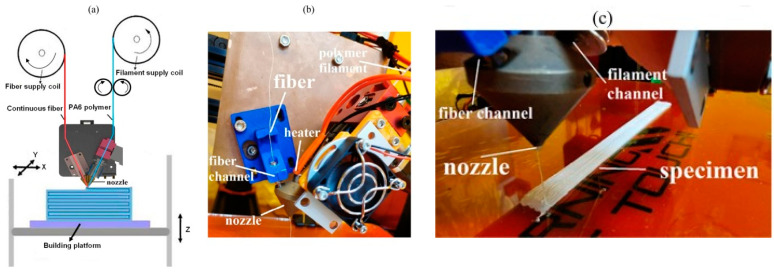
(**a**) Nozzle impregnation of flax with PA6, (**b**) altered printer, and (**c**) flax with PA6 composite printing [54].

**Figure 6 polymers-15-04435-f006:**
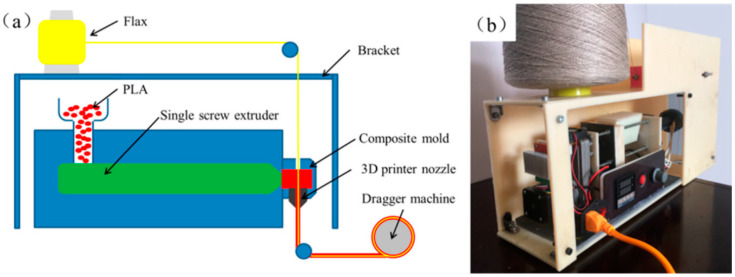
(**a**) CFFRP filament manufacturing representation and (**b**) the CFFRP filament device [58].

**Figure 7 polymers-15-04435-f007:**
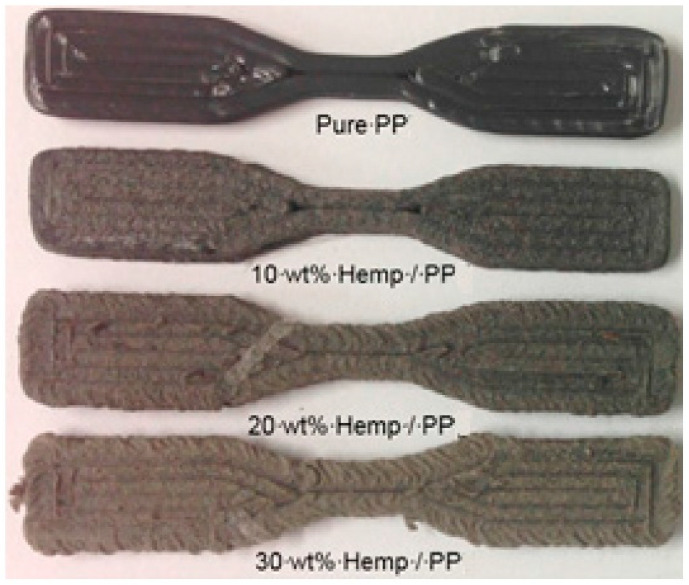
Polypropylene tensile specimen with hemp fiber at various weight proportions [59].

**Figure 8 polymers-15-04435-f008:**
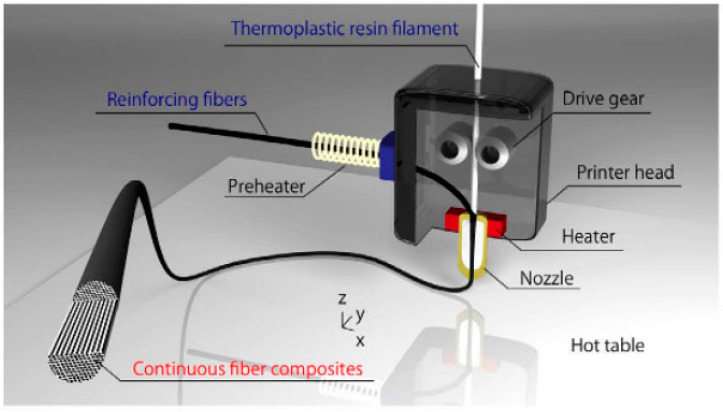
Three-dimensional printing of jute fiber [64].

**Figure 9 polymers-15-04435-f009:**
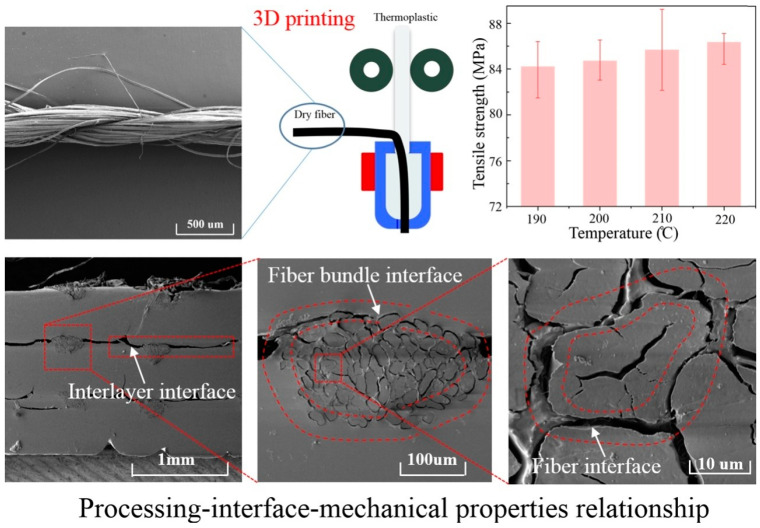
Three-dimensional printing of ramie fiber [65].

**Figure 10 polymers-15-04435-f010:**
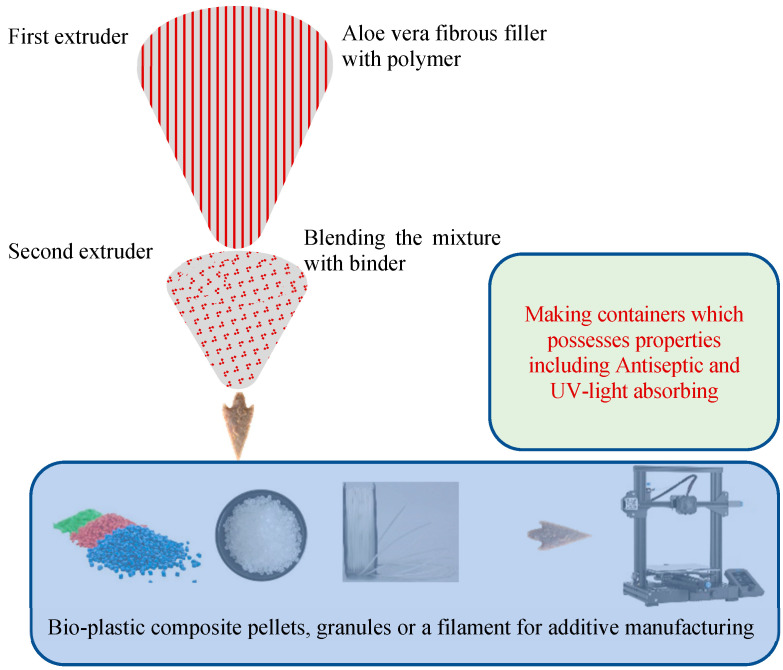
Bioplastic manufacturing.

**Figure 11 polymers-15-04435-f011:**
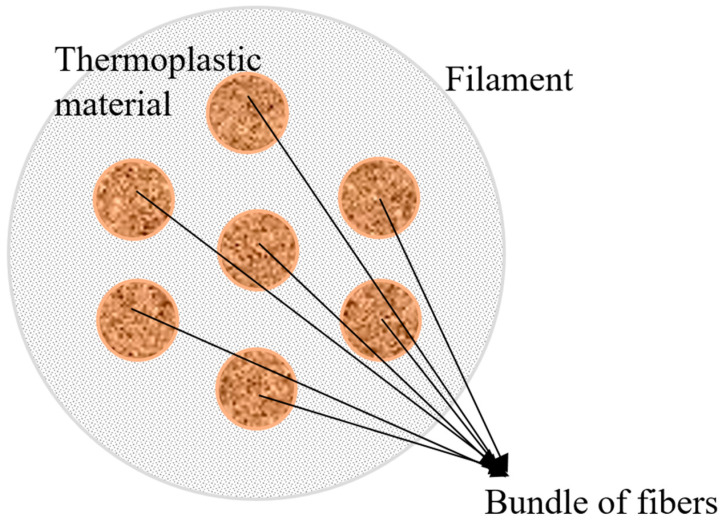
Three-dimensional printing of ramie fiber [71] (redrawn from ref. [71]).

**Table 1 polymers-15-04435-t001:** Mechanical characteristics of 3D printed biocomposites.

S. No.	Composite	Mechanical Property	Ref.
1	ABS with 5% banana fiber	Tensile—4 kgf, Young’s modulus—23	[48]
2	PA6 with 5% banana fiber	Tensile—9.7 kgf, Young’s modulus—127	[48]
3	Thermoplastic with banana fiber	Porosity decreases at a higher printing speed of 0.63% at 320 mm/min	[49]
4	PLA with banana fiber	Tensile on edge build—77%	[50]
5	ABS with 5% kenaf fiber	Tensile—11.5 MPa, Flexural—26.5 MPa	[51]
6	ABS with 10% kenaf fiber	Tensile—18.6 MPa, Flexural—32.6 MPa	[51]
7	PLA with 3 wt. % kenaf fiber	Providing good tensile strength for AFO	[52]
8	PLA with 2.5 wt. % kenaf fiber treated with 6% alkaline and 1% silane	Tensile—57.9 MPa, flexural—84 MPa	[53]
9	PA6 with flax fiber	Better strength in unidirectional composite	[54]
10	PLA with flax fiber	Tensile—50 MPa, bending—73 MPa	[57]
11	PLA with continuous flax fiber-reinforced plastic	Tensile is increased by 89%	[58]
12	PP with 30 wt. % hemp	Tensile is increased by 50%	[59]
13	PLA with 10 wt. % hemp	Increase of 7.3% in ultimate flexural strength	[61]
14	PLA with jute fiber with flame retardant and adhesive	Elongation of 14%	[62]
15	PLA with 5% of waste jute	Improved strength	[63]
16	PLA with jute	Tensile—185 MPa	[64]
17	PLA with ramie	Tensile—86 MPa	[65]
18	PLA with ramie	Maximum penetration force increased by 18%	[66]

**Table 2 polymers-15-04435-t002:** Patent landscape analysis for additive manufacturing of natural fiber [68].

S. No.	Countries	Count	IPC	Count	Year	Count
1	Patent Cooperation Treaty	341	B29C	196	2014	28
2	United States of America	307	B33Y	148	2015	31
3	European Patent Office	17	A61F	91	2016	20
4	India	9	B32B	80	2017	52
5	Canada	4	A43B	42	2018	73
6	South Africa	3	B29K	36	2019	86
7	Australia	1	C09D	36	2020	66
8	Finland	1	C08L	35	2021	82
9	United Kingdom	1	B29L	32	2022	88
10			C08J	32	2023	50

## Data Availability

Not applicable.

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
