# Peer review of "An Overview of the Additive Manufacturing of Bast Fiber-Reinforced Composites and Envisaging Advancements Using the Patent Landscape"

_polymers, 2023, doi:10.3390/polym15224435_

Round 1
Reviewer 1 Report
Comments and Suggestions for Authors
The manuscript reviewed the utilization of additive manufacturing in printing various natural fibers, and the potential landscape of this technique were also listed. This review is interesting, but there are some issues need to be addressed:
1. There are higher proportion of exposition on the civil engineering and soil in the introduction, while these contents are not closely related to the object of this review. It is recommended to delete or simplify this part.
2. Have all types of natural fibers been listed in the manuscript? For instance, the cotton fiber, animal fiber, etc., please add if there is any.
3. The mechanical properties are mentioned in almost all the listed works. The data of mechanical properties are recommended to summarize in a table.
4. The even distribution of the fiber in the matrix is also a critical problem needed to be resolve.
5. The Conclusion part needs to be extended by adding comprehensive discussion on the research trends and suggestions for future works based on the review presented in this manuscript.
6. There are some grammar and typo errors. Please check the manuscript carefully and revise them.
Comments on the Quality of English LanguageOverall, the English language is fine. But there are still some grammar and typo errors. Please check the manuscript carefully and revise them.
Reviewer 2 Report
Comments and Suggestions for Authors
The manuscript relates to the researchers' concern with the various natural fibers 3D printed, and envisaged the future scope of additive manufacturing of natural fiber-reinforced composite using patent landscape. However, this paper is not suitable for publication in this form, which needs to be rewritten in light of the recommendations mentioned below.
Some major concerns are shown as following:
1. Please use three-line form.
2. Please check the references carefully and then unify the format of the references.
3. Some of the captions in Figures 1 and 10 are obscured or missing. Please check and modify them.
4. Please use the international standard system of units. For example, you need to change “percent” to “%”, “weight percent” to “wt%”, and “Degree Celsius” to “°C”. Please check the full text carefully and make changes to similar questions.
5. There are many chemical formulas in the paper that need to change the corresponding numbers to subscripts. For example, you need to change “CO2” to “CO2”. Please check the full text carefully and make changes to similar questions.
6. Abstract: The Abstract section should briefly describe the framework of this review and the significance of writing this review.
7. Introduction: It is necessary to clearly state in this part how this research differs from the research in the literature, that is, the authors should include the novelty of the manuscript.
8. In order to facilitate readers to understand the research field introduced in this paper, and also prove that the additive manufacturing of natural fiber reinforced composites has indeed attracted extensive attention from scholars at home and abroad in recent years, it is suggested that the authors add the trend chart of publications published in this field in the past 20 years. And add the trend chart of publications published about different natural fiber printing methods.
9. In Part 2, when introducing the use of additive manufacturing technology to print various natural fibers, it should not be a simple References list, but should summarize the current results and problems of reinforced composites prepared with different natural fibers, in order to finally make prospects for related research fields.
10. Conclusion: The conclusion should give a brief outlook on the techniques described. In addition, the significance of this article should be expounded.
11. The key performance indicators of reinforced composites prepared with different natural fibers should be summarized in tabular form, and their advantages and disadvantages should be compared.
12. The following two articles are well written and authors are advised to read and cite them.
1) Khan, F.M., Shah, A.H., Wang, S. et al. A Comprehensive Review on Epoxy Biocomposites Based on Natural Fibers and Bio-fillers: Challenges, Recent Developments and Applications. Adv. Fiber Mater. 4, 683–704 (2022). https://doi.org/10.1007/s42765-022-00143-w
2) Luan, P., Zhao, X., Copenhaver, K. et al. Turning Natural Herbaceous Fibers into Advanced Materials for Sustainability. Adv. Fiber Mater. 4, 736–757 (2022). https://doi.org/10.1007/s42765-022-00151-w
Comments on the Quality of English LanguageMinor editing of English language required.
Reviewer 3 Report
Comments and Suggestions for Authors
The paper is an interesting review of the current literature on materials fro 3D printing charged with natural fibers. However, the paper cannot be accepted in its current state.
First of all, the organization of the paper is not really clear: it seems like three main topics were assembled together without a clear main direction. Sections 1 to 3 deals in general with natural charges in polymeric materials; section 4 is on the patent without a clear connection with both the preceding part or the following; section 5 is again on the techniques for natural fibers but again the scope and connections is unclear (does it deal with the manufacturing of the fibers? or mixing and charging polymeric matrices which is also addressed in the first part?).
Additionally it is equally unclear the field of application of these materials: in building only or more in general in construction of mechanical parts or packaging or whatever? The different applications directs towards different polymeric matrices, charges - boith mineral or natural - and manufacturing techniques (the FFF, which is unsuitable for building, is addressed but not only).
Finally, the paper is quite difficult to read and follow because of the writing style which is convoluted, confusing, and heavy. There are long sentences that do not come to a point. There are also many grammatical errors (some are marked in the attached revisions). The style is also quite variable like it is written by several independent hands without a unifying eye on it (for example in some part the "%" symbol is replaced by the express word "percent" - less readable). Is it maybe the result of simply taking those text from different sources without a careful reanalysis and proper rewriting?
Figures and tables are also arguable: the figures, all coming from the references - almost obvious due to the reviewing nature of the manuscript - does not seem very useful for explaining the contents but seem almost randomly chosen just to have some nice picture; tables, or better the only table 1, is unclear and should be better described: the columns are not self-explaining and, so, they must be described; for example there are three columns named "Count" with different numbers but the association is completely puzzling: I must argue that the second column refers to the third and is the number of patents per country (what years interval?) but the 6 and 7 is the total number per year (all the countries). If such, these are three tables not one only (the row number is misleading).

The paper needs a thorough revision or, even better, an almost complete rewriting.
Round 2
Reviewer 1 Report
Comments and Suggestions for Authors
This revised manuscript can be accepted for publication.
Reviewer 3 Report
Comments and Suggestions for Authors
The authors took sufficient care of the comments. Some more could be done (for example table 1 could be improved, modifications are limited to a basic explanation of the columns not really satisfactory).
However, the paper can be accepted possibly with some further adjustments as suggested.
Comments on the Quality of English LanguageThe level of English was a little bit improved and is acceptable.